# Increased risk of slippage upon disengagement of the mitotic checkpoint

**Alma Beatrix Stier**[1☉], **Paolo Bonaiuti**[2☉], **János Juhász**[1], **Fridolin Gross**[3], **Andrea Ciliberto**[1,2]*

**1** Pázmány Péter Catholic University, Faculty of Information Technology and Bionics, Budapest, Hungary, **2** IFOM-ETS, The AIRC Institute of Molecular Oncology, Milan, Italy, **3** Université de Bordeaux, CNRS, ImmunoConcEpT, UMR5164, F-33000, Bordeaux, France

☉ These authors contributed equally to this work.
* andrea.ciliberto@ifom.eu

**Data availability statement:** The code for the numerical simulations is

## Abstract

Drugs that impair microtubule dynamics alter microtubule-kinetochore attachment and invoke the mitotic checkpoint which arrests cells in mitosis. The arrest can last for hours, but it is leaky: cells adapt (i.e., slip out of it) and exit from mitosis. Here, we investigate the mechanism that allows cells to escape, and whether it is possible to prevent it. Based on a model of the mitotic checkpoint which includes the presence of a positive feedback loop, the escape from the arrest is described as a stochastic transition driven by fluctuations of molecular components from a checkpoint ON to a checkpoint OFF state. According to the model, drug removal further facilitates adaptation, a prediction we confirmed in budding yeast. The model suggests two ways to avoid adaptation: inhibition of APC/C and strengthening the mitotic checkpoint. We confirmed experimentally that both alterations decrease the chance of cells slipping out of mitosis, during a prolonged arrest and after washing out the drug. Our results may be relevant for increasing the efficiency of microtubule depolymerizing drugs.

## Author summary

Cell proliferation can be arrested by stimuli that disrupt key events of cell division. To protect themselves from faulty replication, cells in such conditions avoid completing cell division but stall at specific stages of the cell cycle. We investigated the consequences of prolonged but transient stalling in mitosis. We show that cells become prone to undergo errors in chromosome segregation upon transient disruption of microtubule dynamics. Using a combination of mathematical models and experiments, we propose that when cells are stalled in mitosis, stochastic fluctuation spontaneously drive cells out of the arrest. Such fluctuations are more disruptive when the effect of the drug fades off. Based on this knowledge, we show ways to allow cells to escape from stalling without compromising their genetic integrity. The result is relevant for experimental biologists, who often drisupt microtubules transiently as a 'neutral' way of synchronizing cells. It is also relevant for patients treatment, since microtubules destabilizers are often used in the clinics.

available at: https://github.com/sbeatrix/WIA_source_codes/tree/main

**Funding:** This work was supported by the italian association for cancer research (AIRC IG grant 28821 to AC); by donations of the Suma-Nesi family; by the Hungarian National Research, Development and Innovation Office (grant TKP2021-EGA-42 to AC); by the unkp-23-1 new national excellence program of the hungarian ministry for culture and innovation, from the source of the national research, development and innovation fund (UNKP-1-I-PPKE-108 to ABS). The funders had no role in study design, data collection and analysis, decision to publish, or preparation of the manuscript.

**Competing interests:** The authors have declared that no competing interests exist

## Introduction

During mitosis, sister chromatids are segregated to the two daughter cells. This process requires the interaction between specialized chromosomal structures (kinetochores) and polymers of alpha- and beta-tubulin called microtubules. When all chromosomes are attached to microtubules originated by the opposite spindle-pole bodies (centrosomes in metazoans), they can be separated and brought towards the future daughter cells. This process needs to be carefully regulated, to make sure the daughter cells receive the identical genetic material from their mother. Failure in chromosome segregation results in aneuploidy, which comes with a strong fitness cost, primarily related with the proteoxic stress, and is a typical hallmark of cancer cells [1].

A surveillance mechanism known as the mitotic checkpoint supervises the process of chromosome segregation and arrests cell cycle progression when the conditions for a faultless separation of sister chromatids are not met [2]. The checkpoint inhibits an E3 ubiquitin ligase, the anaphase promoting complex or cyclosome (APC/C) which targets for degradation mitotic cyclins and securin (in budding yeast: Clb2 and Pds1, respectively). The molecular target of the checkpoint in fact is Cdc20, a co-factor of APC/C. Hence, when kinetochores are not attached to microtubules, or when there is no tension between sister chromatids, the mitotic checkpoint is operational, and leads to the production of the mitotic checkpoint complex (MCC), which includes Mad2, Mad3 and Bub3 as well as Cdc20. APC/C$^{Cdc20}$ bound to MCC cannot target its substrates, and cells cannot transit into anaphase.

Drugs used for cancer treatment (e.g., taxol and vinca alkaloids such as nocodazole) arrest cell proliferation by invoking the mitotic checkpoint. To do so, they impair microtubule dynamics and consequently microtubule-kinetochore attachment. Cells can withstand for several hours a mitotic arrest induced by stimuli that impair microtubule-kinetochore attachment. However, on the long term they escape from the arrest via a process called adaptation or slippage whereby cells enter anaphase with unattached kinetochores [3–6]. Since cells progress into anaphase when microtubules and kinetochores are not properly attached, adaptation is at risk of eliciting chromosome missegregation and aneuploid progeny. The reduced effect of microtubule-targeting drugs is also due to cells' ability to overcome the arrest. Hence, it is important to devise strategies that minimize the chance of cells from adapting. Usually, the problem of adaptation or slippage is analyzed in the constant presence of drugs activating the mitotic checkpoint [3,4,7]. However, adaptation to microtubule targeting drugs should be studied also when the effect of the drug fades off, as it happens during normal treatment.

How adaptation takes place is still debated. It was originally shown that human cells escape from the arrest with an active mitotic checkpoint, as identified by the localization of Mad2 at kinetochores [4]. This result was later confirmed in yeast [3]. Later work suggested the presence of pathways that trigger adaptation during protracted arrest [8]. Performing single-cell analysis in budding yeast, however, we observed that the probability of cells to escape does not change considerably for several hours [3]. Hence, we proposed a model which explains adaptation as a random process driven by the incomplete inactivation of APC/C$^{Cdc20}$, and fueled by the intrinsic noise originated by the small copy number of key molecular players. According to the model, when fluctuations in APC/C$^{Cdc20}$ cross an arbitrary threshold, the latter drives cells into anaphase [3]. However, in that work we did not provide a rationale for the presence of a threshold.

A bistable system (i.e., a system with two co-existing states: a checkpoint ON and a checkpoint OFF state) could provide a rationale for such threshold. Adaptation could occur when random fluctuations drive a system originally in the checkpoint ON state into the basin of attraction of the checkpoint OFF state. Interestingly, several groups have shown that indeed the mitotic checkpoint has properties of a bistable system, and specifically that the mitotic

checkpoint, normally silenced after metaphase, can also be activated in anaphase under non-physiological conditions. This was shown for (i) Mps1 overexpression in budding yeast [9], (ii) stabilization of mitotic cyclins in mammals and fission yeast [10,11], (iii) laser-induced rupture of microtubule-kinetochore attachment in mammalian cells [12], (iv) inactivation of CDK1-counteracting phosphatases in budding yeast [13], (v) prolonged securin degradation in fission yeast [10].

Bistability requires the presence of a positive feedback loop whereby the mitotic checkpoint can sustain its own activity. Several positive feedback loops have been proposed, following two general schemes. One of them includes the double inhibition between APC/C and the mitotic checkpoint [14] (Fig 1A, left). The inhibition that the mitotic checkpoint exerts on APC/C via MCC is straightforward. Less clear is the inhibition that APC/C exerts on the checkpoint. Since components of MCC are not direct targets of APC/C, the relationship must involve other components, more or less directly related with the checkpoint. Mitotic cyclins play a key role in checkpoint activity in several organisms and are degraded by APC/C$^{Cdc20}$, thus being natural candidates for closing the loop [11,15]. However, in budding yeast, the role of mitotic

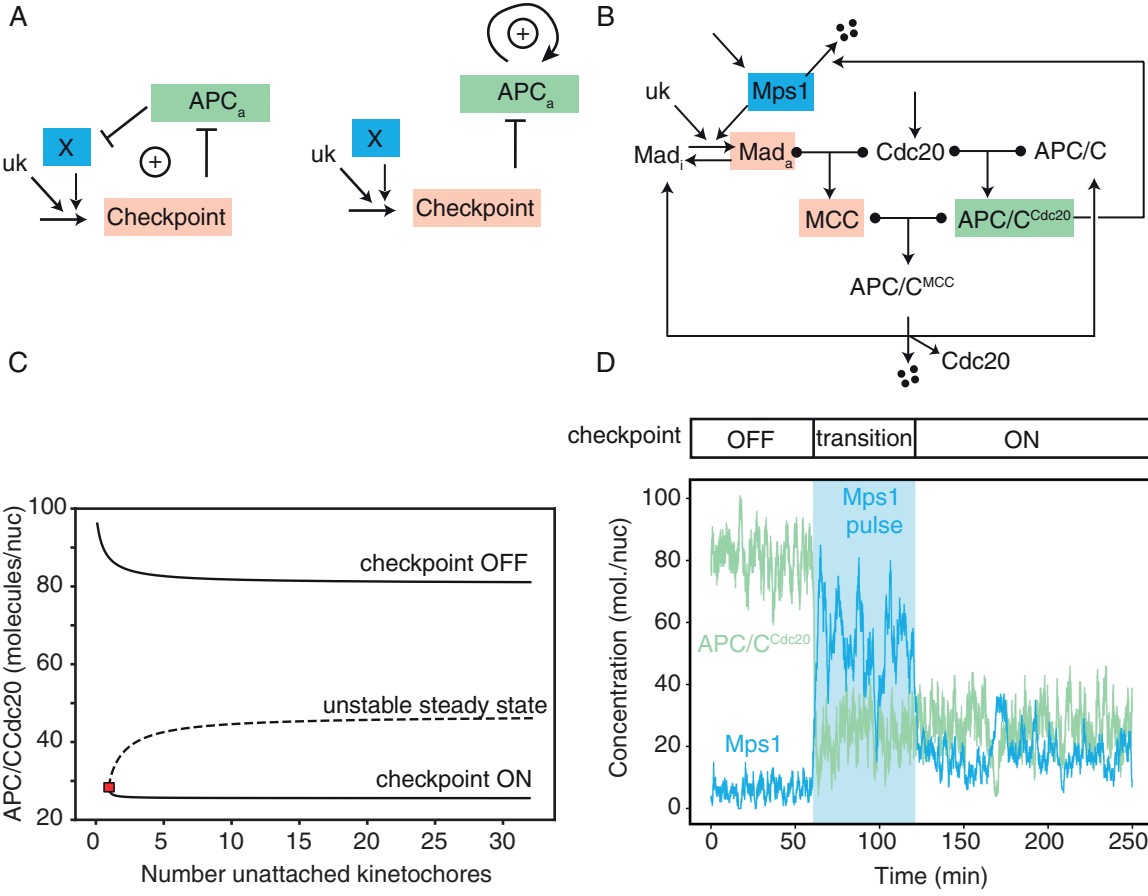

**Fig 1. Detailed model of the mitotic checkpoint with bistability.** **A)** Generic wiring diagrams for positive feedback loops in the mitotic checkpoint network. **B)** Detailed wiring diagram of the mitotic checkpoint. **C)** Bifurcation diagram of the detailed model (wiring in panel **B**). The red square indicates the saddle node bifurcation: the value of unattached kinetochores where the checkpoint ON steady state is created/disappears. In wild type cells, it is located just above 1 unattached kinetochore (nuk=0.9). Equations in S1 Text, and parameters in Table 2. **D)** Stochastic simulations of the Palframans's experiments [9]. Reactions in Table 3, parameters in Table 2, initial conditions in Table 1 (checkpoint OFF). Mps1 overexpression is simulated increasing 3X the synthesis of Mps1 for one hour.

cyclins in supporting checkpoint activity is less clear than in other organisms [6]. Instead, a positive feedback loop was proposed to include Mps1, a member of the mitotic checkpoint degraded by APC/C [9]. A more indirect inhibition was suggested based on the fact that APC/C$^{Cdc20}$ degrades securin which inhibits separase which in turn activates PP2A$^{Cdc55}$: the latter closes the loop inhibiting APC/C$^{Cdc20}$ formation in anaphase [6]. Given the uncertain molecular nature of the network, in the wiring diagram of Fig 1A we include a generic 'X' molecule, which could be, for example, Mps1 or Clb2/Cdk1. A second, entirely different, type of positive feedback loop was proposed under the assumption that APC/C$^{Cdc20}$ induces its own formation by activating the inhibited form of APC/C, Fig 1A right [15,16]. Finally, in mitosis other positive feedback loops (PFLs) are present (e.g., for the synthesis of Cyclin B, double inhibition between Cdk1 and Cdh1 [17]), but they do not include the checkpoint, and as such cannot give rise to two stable steady states (checkpoint ON and OFF).

In this paper, we ask how positive feedback and bistability can help explaining adaptation, and we investigate mechanisms that may prevent cells from adapting thus reducing the risk of missegregation. We show that when the checkpoint is forcibly activated, cells are trapped in a condition that favours entry in anaphase with unattached kinetochores, and even more so when the drug is washed out. Experiments suggested by our model show that fortifying the mitotic checkpoint and decreasing APC/C$^{Cdc20}$ activity are two possible strategies to avoid adaptation during washout.

## Results

### A detailed model of the mitotic checkpoint shows bistability

To explore the behavior of the checkpoint as a bistable system, we reproduced experiments performed for this purpose in budding yeast by Palframan et al [9] who showed that Mps1 creates a positive feedback loop with the mitotic checkpoint, playing the role of X in Fig 1A. Hence, to reproduce their data we modified our previous model [3] by including Mps1 in lieu of X (from now on: the 'detailed model'). In particular, we added the APC/C$^{Cdc20}$-mediated degradation of Mps1, and the Mps1-dependent checkpoint activation (Fig 1B) [9]. The model's details are given in S1 Text Model, where we provide a full description of the equations, including the choice of parameter values which is largely based on available data (molecular species in Table 1, parameters in Table 2).

Bifurcation analysis shows that the model is bistable (Fig 1C). For a number of unattached kinetochores larger than one (or for lack of tension between sister chromatids), cells can either be in a checkpoint ON state (low APC/C$^{Cdc20}$, high Mps1), or in an OFF state (high APC/C$^{Cdc20}$, low Mps1). This analysis was performed with a deterministic model. Furthermore, we developed a stochastic version of the model to investigate whether random fluctuations could enable spontaneous transitions between the steady states. We found that the intrinsic noise supplemented with transcriptional bursts, which are well documented in yeast [18], suffice to produce spontaneous transitions from the checkpoint ON to the checkpoint OFF state (but not vice versa).

We then aimed at reproducing experimental data showing that the checkpoint ON state is present also in cells arrested in anaphase although normally they are not attracted by it. To show the presence of such a checkpoint ON steady state in anaphase, Palframan et al overexpressed Mps1 in this cell cycle phase and observed the engagement of the checkpoint (i.e., Pds1 was stable and cells were trapped in anaphase, Figure 2B-2D in [9]). In this setting, it was the lack of tension that triggered the pathway, all kinetochores being attached. In this context, Mps1 may phosphorylate checkpoint proteins as well as specific residues on kinetochores, given its role both in error correction and checkpoint activation [19]. Remarkably, when the

**Table 1.** Species and initial conditions of used in the model to simulate a checkpoint arrest.

| Species | Name in model | Checkpoint ON (nM) | Checkpoint ON (mol./nucleus) | Checkpoint OFF (nM) | Checkpoint OFF (mol./nucleus) |
|---|---|---|---|---|---|
| free Cdc20 | $C$ | 5.2 | 13 | 176 | 440 |
| free APC | $A$ | 18 | 45 | 0.8 | 2 |
| APC/C$^{Cdc20}$ | $AC$ | 10 | 25 | 32.8 | 82 |
| Mad (active) | $Mad_a$ | 56 | 140 | 0.8 | 2 |
| Mad (inactive) | $Mad_i$ | 0.8 | 2 | 62.4 | 156 |
| Mps1 | $Mps1$ | 6.8 | 17 | 2.4 | 6 |
| Free MCC | $MC$ | 1.6 | 4 | 0 | 0 |
| APC/C$^{MCC}$ | $ACMC$ | 11.6 | 29 | 6.4 | 16 |
| APC total | $Atot$ | 40 | 100 | 40 | 100 |
| Mad total | $Mtot$ | 70 | 175 | 70 | 175 |
| Cdc20 total | $Ctot$ | 40 | 100 | 222 | 555 |

ectopic overexpression was switched off, Mps1 and Pds1 -- targets of APC/C$^{Cdc20}$ -- remained stable, showing that the checkpoint remained operational in anaphase-arrested cells (Figures S4 and 4B in [9]).

This result is a formal proof of bistability, since cells are in two different states (checkpoint ON/OFF) in identical experimental conditions (anaphase arrest). The model well recapitulates these results. As shown by bifurcation diagrams (S1 Fig), the overexpression of Mps1 shrinks the checkpoint OFF state so that the system is attracted to the ON state. When Mps1 overexpression is switched off, however, cells remain trapped in the checkpoint ON state. One typical trajectory is shown in Fig 1D.

We conclude that the detailed model can reproduce data obtained in budding yeast which proved the mitotic checkpoint to be a bistable system.

## The detailed model reproduces experimental adaptation dynamics

Having reproduced key experiments showing the bistable nature of the mitotic checkpoint in anaphase, we used the model to reproduce the observed pattern of adaptation in the constant presence of drugs. Adaptation was studied by expressing GFP-tagged Mad2 and mCherry-tagged Clb2. Cells were arrested in microfluidic devices by α-factor and released in nocodazole, a drug the depolymerizes microtubules. This setting allowed us to observe individual cells for prolonged time upon drug-induced microtubule depolymerization. Localization of Mad2 was used as a read-out for checkpoint signaling, and Clb2 degradation for APC/C$^{Cdc20}$ activation (i.e., localized Mad2 implies checkpoint activation while Clb2 degradation implies APC/C$^{Cdc20}$ activity).

The large majority of cells (96.9%) eventually degraded Clb2 (S2A Fig), indicating progression into anaphase. Following the protocol adopted in [3], cells that degraded Clb2 with Mad2 localized, and thus in the presence of an active checkpoint, were scored as undergoing adaptation. Cells that did so after Mad2 delocalization were classified as cells that satisfied the checkpoint (a schematic representation of these definitions is in Fig 2A and real examples in S2B Fig). In agreement with previous results [3], the cumulative distribution of adaptation times -- the time elapsed between Cyclin B accumulation and degradation -- followed an exponential distribution after a delay (Fig 2B). The delay can be interpreted as the time that elapses between cyclin B accumulation and the formation of active APC/C$^{Cdc20}$.

**Table 2. Parameters used in the detailed model for WT simulations. Choice of parameters is given in S1 Text.**

| Description | Parameter | Value (deterministic) | Value (stochastic) | Reference |
|---|---|---|---|---|
| Cdc20 degradation rate | $k_{deg}$ | 1.3E0 min⁻¹ | 1.3E0 min⁻¹ | constrained by Cdc20 degradation kinetics [3] |
| Cdc20 synthesis rate | $k_{sync20}$ | 1.7E+1 nM x min⁻¹ | 4.2E1 mol./nuc x min⁻¹ | fitted to have total Cdc20 40 nM in wild-type cells |
| MCC formation rate | $k_{assMC}$ | 8.8E-3 (nM x min)⁻¹ | 2.2E-2 (mol./nuc x min)⁻¹ | fitted to have free MC 2 nM |
| MCC dissociation rate | $kD_{MC}$ | 5.6E-1 nM | 1.4E0 mol./nuc | kDMC = kdissmc/kassmc |
| APC/C^Cdc20 formation rate | $k_{assAC}$ | 4.4E-2 (nM x min)⁻¹ | 1.1E-1 (mol./nuc x min)⁻¹ | fitted to have APCtot 40 nM |
| APC/C^Cdc20 dissociation rate | $kD_{AC}$ | 3.4E0 nM | 8.5E0 mol/nuc | kDAC = kdissmc/kassmc |
| APC/C^MCC formation rate | $k_{assACMC}$ | 2.3E-1 (nM x min)⁻¹ | 5.8E-1 (mol./nuc x min)⁻¹ | fitted to have APC/C^MCC 12 nM in wildtype checkpoint ON |
| APC/C^MCC dissociation rate | $kD_{ACMC}$ | 4.4E-1 nM | 1.1E0 mol./nuc | kDACMC = kdissacmc/kassacmc |
| Background degradation rate for Cdc20, MCC, APC/C^Cdc20, APC/C^MCC | $k_{degBG}$ | 3.8E-2 min⁻¹ | 3.8E-2 min⁻¹ | fitted to be ~10x smaller than main degradation |
| Activation rate of Mad | $k_{act}$ | 2.4E-2 min⁻¹ | 2.4E-2 min⁻¹ | fitted to reproduce the measured concentration of 70 nM |
| Inactivation rate of Mad | $k_{inact}$ | 2.7E-3(nM x min)⁻¹ | 6.7E-3(mol./nuc x min)⁻¹ | |
| Mps1 synthesis rate | $k_{synX}$ | 4.4E0 (nM x min)⁻¹ | 1.1E1 (mol./nuc x min)⁻¹ | fitted to have total X (Mps1) 40 mol./cell in wild-type cells |
| Mps1 degradation rate | $k_{degX}$ | 9.6E-3 (nM x min)⁻¹ | 2.4E-2 (mol./nuc x min)⁻¹ | fitted to reproduce the degradation dynamics of [9] |
| Mps1 background degradation rate | $k_{degBGX}$ | 6.2E-2 min⁻¹ | 6.2E-2 min⁻¹ | fitted to be ~10x smaller than main degradation |
| Kinetochore attachment rate | $k_{att}$ | 3.0E-2 min⁻¹ | 3.0E-2 min⁻¹ | fitted to simulate drug washout time interval (S3B Fig) |
| Saturation function for nUK signal | $k'$ | 1.7E+2 | 1.7E+2 | from [29] |
| Saturation function for nUK signal | $J_n$ | 4.0E-1 | 4.0E-1 | from [29] |
| Michaelis-Menten constant for Mad activation | $J$ | 0.84E0 nM | 2.1E0 mol./nuc | – |
| Bursting parameter switching the ON state for Cdc20 | $k_{onCDC20}$ | – | 1.0E0 | – |
| Bursting parameter switching the OFF state for Cdc20 | $k_{offCDC20}$ | – | 1.0E-1 | – |
| Bursting parameter switching the ON state for Mps1 | $k_{onX}$ | – | 1.0E0 | – |
| Bursting parameter switching the OFF state Mps1 | $k_{offX}$ | – | 1.0E-1 | – |

We then set out to reproduce these results with the model. We can use the bifurcation diagram to understand the meaning of checkpoint satisfaction and adaptation in the model (Fig 2C). When the checkpoint is satisfied, APC/C^Cdc20 increases after the attachment of the last unattached kinetochore. This last attachment leads to the loss of the checkpoint ON state and thereby pushes the system to the checkpoint OFF state in an all-or-none transition. For this, it is necessary that the left border of the bistability region (corresponding to a saddle-node bifurcation, red square in Fig 1C) is located between 0 and 1 unattached kinetochores. Adaptation, instead, occurs when random fluctuations drive a system originally in the checkpoint ON state into the basin of attraction of the checkpoint OFF state when not all kinetochores have attached. The checkpoint is thereby silenced in the presence of unattached kinetochores.

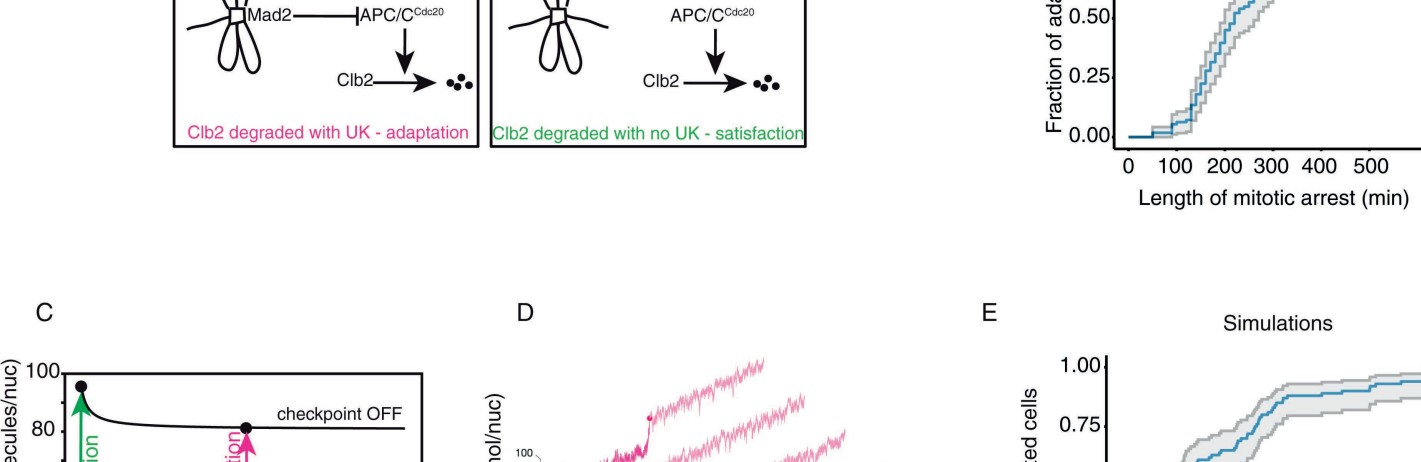

**Fig 2. Adaptation dynamics with constant unattached kinetochores. A)** Definition of adaptation in the experimental system. UK stands for unattached kineto-chores. **B)** Cumulative distribution of mitotic arrest length. Cells growing in YPD are arrested in G1 and released in nocodazole, indefinitely. Cells carry Mad2-GFP to record checkpoint activation, and Clb2-mCherry to record both mitotic entry (Clb2 rise time) and mitotic exit (Clb2 degradation time). Length of mitotic arrest is defined as the difference between mitotic exit and mitotic entry. **C)** Definition of adaptation in the model. Adaptation occurs when the system transits from the ON to the OFF state with unattached kinetochores. **D)** Stochastic simulations of APC/C$^{Cdc20}$ dynamics with fixed number of unattached kinetochores (nuk=10). Reactions in [Table 3](), parameters in [Table 2](), initial conditions in [Table 1]() (checkpoint ON). Filled circles indicate the time APC/C$^{Cdc20}$ crosses the activation threshold (80 mol/cell), after which we assume entry into anaphase. **E)** Cumulative distribution of simulated mitotic arrests, computed as the difference between the start of the simulation and the time APC/C$^{Cdc20}$ crosses the activation threshold. Representative trajectories are shown in panel **(D)**.

We started stochastic simulations from the checkpoint ON state with a fixed number of unattached kinetochores. With time, we observed that fluctuations in protein levels managed to drive the system to the checkpoint OFF state in the presence of unattached kinetochores ([Fig 2D]()). When we produced the cumulative distribution of adaptation times, we observed that within ~600 minutes all trajectories adapted, showing an exponential distribution similar to that observed experimentally, [Fig 2E](). In simulations, we do not observe a delay since simulations start with already formed APC/C$^{Cdc20}$.

We conclude that a stochastic version of our model with the addition of the positive feedback based on Mps1 accounts for the experimental adaptation dynamics observed in budding yeast.

## Washing out microtubule targeting drugs triggers adaptation

We then asked whether a similar behavior was observed upon transient destabilization of microtubules, and thus transient engagement of the mitotic checkpoint. We performed a wash-out experiment using again cells carrying GFP-tagged Mad2 and mCherry-tagged Clb2, to distinguish between adaptation and checkpoint satisfaction. We followed the same protocol

described previously, but this time the drug was washed out three hours after the arrest. As the drug is washed out, its effect fades off, and microtubules are able again to polymerize and interact with kinetochores. As before, we followed the interplay of Mad2 localization and Clb2 degradation to distinguish between checkpoint satisfaction and adaptation (S1 and S2 Movies, respectively). The majority of cells (60.8%) started Clb2 degradation in the presence of localized Mad2 even after the washout. Instead, 39.2% degraded Clb2 after satisfying the checkpoint, with Mad2 delocalized (Figs 3A, left, and S3A).

We asked whether the model would reproduce this behavior during washout. We described microtubule-kinetochore attachment as a stochastic process in which the rate of attachment is proportional to the remaining number unattached kinetochores. To set the rate parameter, we used the experimentally measured rate of Mad2 removal from kinetochores as a proxy for kinetochore attachment, S3B Fig. Again, adaptation is defined as a transition towards the OFF state in the presence of unattached kinetochores (nUK>0), whereas transition with nUK = 0 means that the checkpoint is satisfied (Fig 2C). We then ran stochastic simulations and kept track of adaptation and checkpoint satisfaction. In agreement with experimental data, a large fraction of trajectories slipped through the arrest when we allowed kinetochores to attach (Fig 3A right, individual trajectories in Fig 3B).

We used the model to investigate the molecular mechanism underlying adaptation during washout. Cells that satisfied the checkpoint were attracted to the checkpoint ON state until the last unattached kinetochore (S3C Fig left). Early adapters (those adapting before ~50 minutes) managed to escape when attachment had just started and so they adapted with many unattached kinetochores (S3C Fig, right). As the number of unattached kinetochores decreased, the distance between stable and unstable steady state (proportional to the checkpoint strength) decreased as well. Hence, random fluctuations (here shown in a 2D projection) that may have not sufficed to escape the arrest with many unattached kinetochores were enough to drive cells in the checkpoint OFF state with few unattached kinetochores (S3C Fig, center). These late adapters were more numerous and escaped with few unattached kinetochores (Fig 3C left, upper panel). The presence of early adapters shifted the distribution of adapting cells to earlier time points compared to the distribution of cells satisfying the checkpoint (Fig 3C left, lower panel).

These results were confirmed in the experiments, where the distribution of adapting cells was anticipated compared to cells satisfying the checkpoint (Fig 3C right, lower panel). As a proxy for the number of unattached kinetochores, we measured the amount of Mad2 at the time of adaptation. In agreement with the model, it correlated negatively with adaptation times: cells that adapted earlier had more Mad2 signal and thus likely more unattached kinetochores (Fig 3C right, upper panel).

In summary, both simulations and experiments showed that more than half of the cells activated APC/C$^{Cdc20}$ when Mad2 was localized upon nocodazole washout. Timing of APC/C$^{Cdc20}$ activation was negatively correlated with the extent of Mad2 localization.

## Preventing adaptation with APC/C mutants during washout

According to the model, to prevent adaptation we need to minimize the effect of fluctuations that drive cells from the checkpoint ON to the checkpoint OFF state. For that, it is necessary to increase the distance between the checkpoint ON and the unstable steady state that separates it from the checkpoint OFF state (S4A Fig). This can be achieved, for example, by decreasing the affinity of APC/C for Cdc20 (Fig 4A).

Such a mutant does exist: it is a mutant of APC/C (we call it APC-A) partially impaired in binding Cdc20. The defect is due to the mutation to alanine of all serine/proline (SP) and threonine/proline sites (TP) on two APC/C subunits: Cdc16 (Apc6 in mammals) and Cdc27

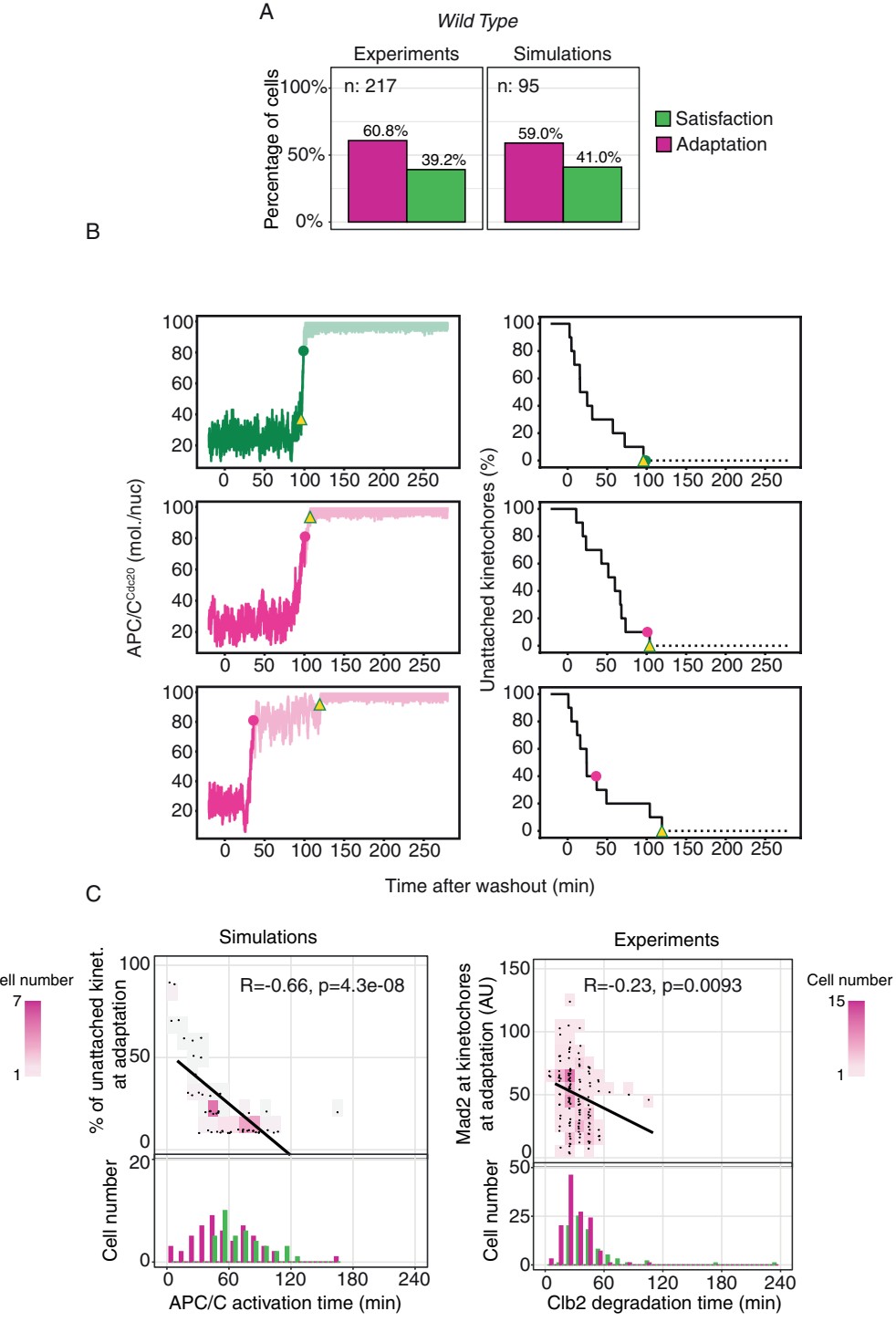

**Fig 3. Adaptation following drug washout. A)** Left panel: Wild-type cells growing in YPRG are arrested in G1 and released in nocodazole for 180 minutes, after which nocodazole is removed from the media. Cells carry Mad2-GFP to record checkpoint activation, and Clb2-mCherry to record mitotic exit (Clb2 degradation time). Adaptation and exit are defined based on Mad2 localization at Clb2 degradation time (see Figs 2A and S2B). Only cells properly degrading Clb2 are included in the percentage (see S2A Fig). Right panel: same percentages for the simulated data, where adaptation is defined as APC/C activation with at least one unattached kinetochore (see Fig 2A). **B)** Representative stochastic simulations of APC/C$^{Cdc20}$ dynamics (left) from the time kinetochores are allowed to attach (right). Yellow triangles indicate the moment when all kinetochores have attached; filled circles mark the time APC/C$^{Cdc20}$ crosses the activation threshold. Reactions in Table 3, parameters in Table 2, initial conditions in Table 1 (checkpoint ON).

**C)** Left panel, top: scatter plot of the number of unattached kinetochores at APC/C activation in wild-type adapting cells. Black line shows the linear regression, R- and p-values from Pearson's correlation. Bottom: distribution of APC/C$^{Cdc20}$ activation time in cells adapting (purple) or exiting (green). Data from 100 simulations as in panel (**B**). Right panel, top: scatter plot of the amount of Mad2 at kinetochores (see Material and Methods) in wild-type adapting cells. The black line shows the linear regression, R- and p-values from Pearson's correlation. Bottom: histogram of Clb2 degradation time in adapting (purple) or exiting cells (green).

(Apc3 in mammals) [20].To quantify the changes introduced by the mutant, we performed fluorescence cross correlation spectroscopy (FCSS) experiments. This technique allows to measure absolute values of proteins or protein complexes tagged with fluorescent markers. Using Mad3-mCherry and Mad2-GFP we measured MCC levels during an arrest, while Mad2-GFP and Cdc23-mCherry allowed us to follow APC/C$^{MCC}$. Results show an increase of MCC in the APC-A mutant, whereas APC/C$^{MCC}$ does not change significantly (S4B Fig).

For the adaptation experiments, cells expressing Mad2-GFP and Clb2-mCherry and carrying APC-A mutations were synchronized and arrested in nocodazole. In the constant presence of nocodazole, most cells kept the mitotic arrest and adaptation was largely reduced (S4C Fig, left and central panels). During washout, compared to the wild type, we observed a decrease in the fraction of adapting cells (compare Fig 4B left with Fig 3A left). It is worth noticing that even upon washout a sizeable fraction of APC-A cells (11.4%) failed to degrade Clb2 (S4D Fig).

We used the model to interpret these data. We introduced the reduction in APC/C - Cdc20 binding based on the observed changes in APC/C$^{MCC}$ and MCC (bifurcation diagram in S4E Fig), and performed simulations in the constant presence of unattached kinetochores. As expected, adaptation was decreased to less than 20% (Fig 4B, right panel). When we simulated nocodazole washout, we observed that adaptation occurred primarily when fluctuations were within reach of the unstable steady state, and so close to the saddle node. Here, unavoidable physiological fluctuations forced adaptation, similarly to late adaptations in wild type cells (S3C Fig center). This led to the disappearance of early adapters, a result which came with three main consequences, all in agreement with the experimental results: the fraction of adapting cells decreased (Fig 4B), adapting cells and cells satisfying the checkpoint had similar distributions (Fig 4C, upper panel) and on average APC-A cells took longer to exit mitosis (i.e., to degrade Clb2) than wild types (Fig 4D).

In conclusion, our model suggests that a mutant impaired in APC/C$^{Cdc20}$ binding would decrease the instances of adaptation upon drug washout. Experimentally, we confirmed this result using an APC/C mutant impaired in Cdc20 binding showing that it prevents adaptation during wash-out, limiting it to cells that carry a small number of unattached kinetochores.

## Preventing adaptation with Mad2 overexpression after washout

Inhibiting APC/C does prevent adaptation, but also arrests many cells in mitosis even after washing out the drug (S4D Fig). Thus, we looked for an alternative approach to decrease the effect of random fluctuations.

According to the model, reinforcing the mitotic checkpoint by overexpressing Mad2 would separate checkpoint ON and unstable steady state similarly to what was observed with APC-A (Fig 5A). Mad2 overexpression can be experimentally obtained using cells expressing it from one copy of the galactose-inducible *GAL1* promoter, which produces a level of overexpression unable on its own to alter cell cycle dynamics [21]. To simulate this mutant, in the model we increased the total amount of Mad2 with aim of reproducing the reduced adaptation of *GAL1-MAD2* cells observed experimentally in the constant presence of nocodazole (S5A Fig). With this optimized level of overexpression, we obtained a bifurcation diagram similar to that we produced for APC-A (S5B Fig).

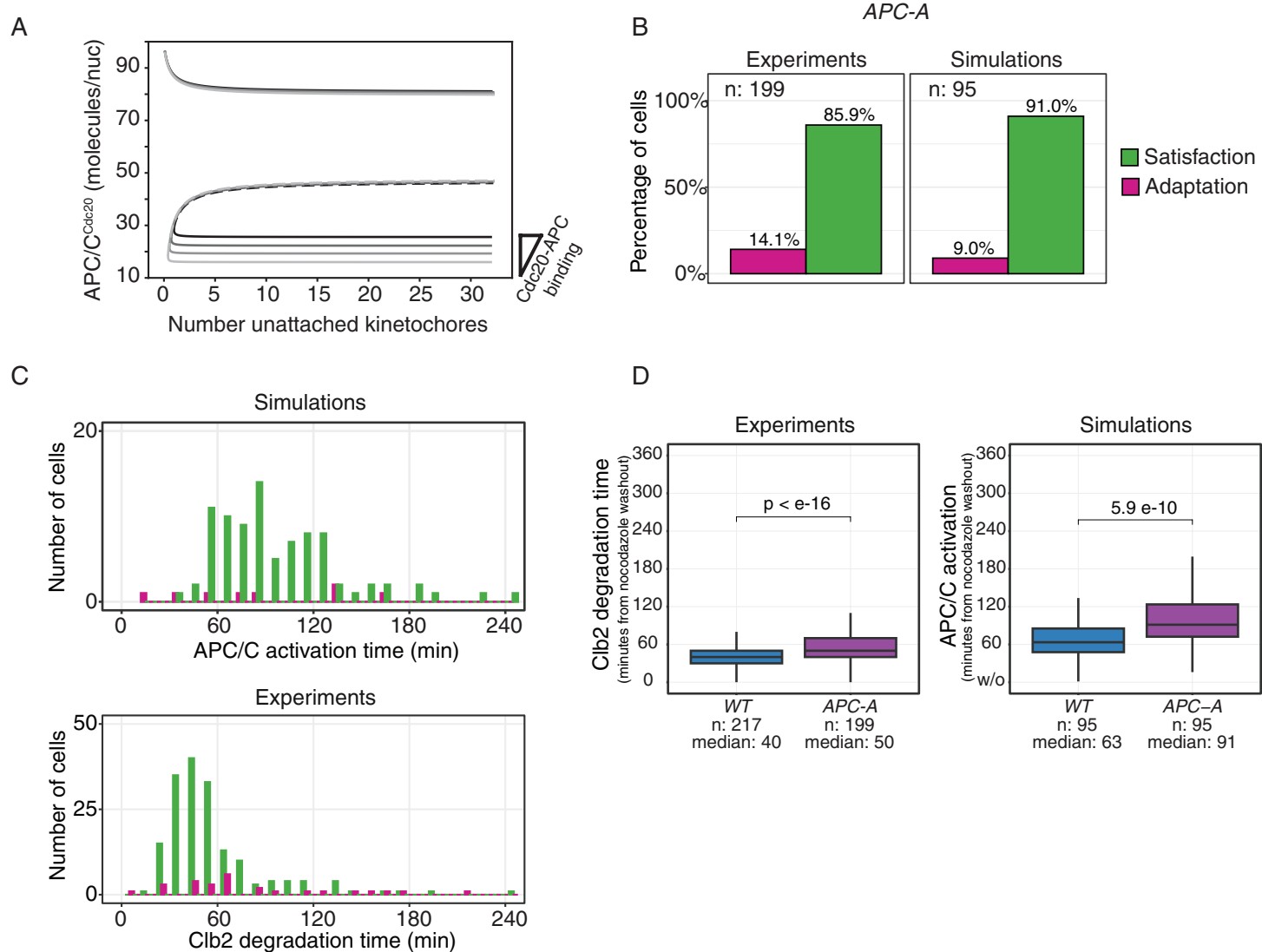

**Fig 4. Dynamics of nocodazole washout in APC-A mutants. A)** Bifurcation diagrams showing the effect of decreasing APC/C$^{Cdc20}$ binding to 88%, 79% and 69% of Wild Type levels. Wild type in black. Equations in S1 Text, and parameters in Table 2. **B)** Adapting vs exiting cells. Left panel: APC-A cells growing in YPRG were arrested in G1 and released in nocodazole for 180 minutes, after which nocodazole was removed from the media. Cells carry Mad2-GFP to record mitotic checkpoint activation, and Clb2-mCherry to record mitotic exit (Clb2 degradation time). Adaptation and exit are defined based on Mad2 localization at Clb2 degradation time (see S2B Fig). Only cells degrading completely Clb2 are included in the percentage (see S2A Fig). Right panel: same percentages for the simulated data, where adaptation is defined as APC/C reaching the activation threshold with at least one unattached kinetochore. Reactions in Table 3, parameters in Table 2, initial conditions in Table 1 (checkpoint ON). At time 0, kinetochore attachment is allowed. **C)** Different dynamics of cells exiting and adapting. Upper panel: histograms of APC/C activation time in in cells adapting (purple) or exiting properly (green). Simulations details as in panel **(B)** right. Lower panel: histograms of Clb2 degradation time in in cells adapting (purple) or exiting (green). Experimental details as in **(B)** left. **D)** Comparison of Clb2 degradation time (experiments) and APC/C activation (simulations) in wild-type and APC-A cells. Left, experimental data: cells are the same presented in the barplots in Figs 3A and 4B. Right, results from wash-out simulations. Equations in S1 Text, and parameters in Table 2.

When we simulated cells overexpressing Mad2 levels upon drug washout, again the fraction of cells slipping from the arrest was decreased (compare Fig 5B with Fig 3A, right). In the same simulation, similarly to the simulated APC-A, we observed a decrease of adaptation with multiple unattached kinetochores and no difference in the distribution of cells adapting and satisfying the checkpoint (Fig 5C). Also Mad2 overexpression prevented early adapters and so it also delayed the average time of adaptation (Fig 5D, left).

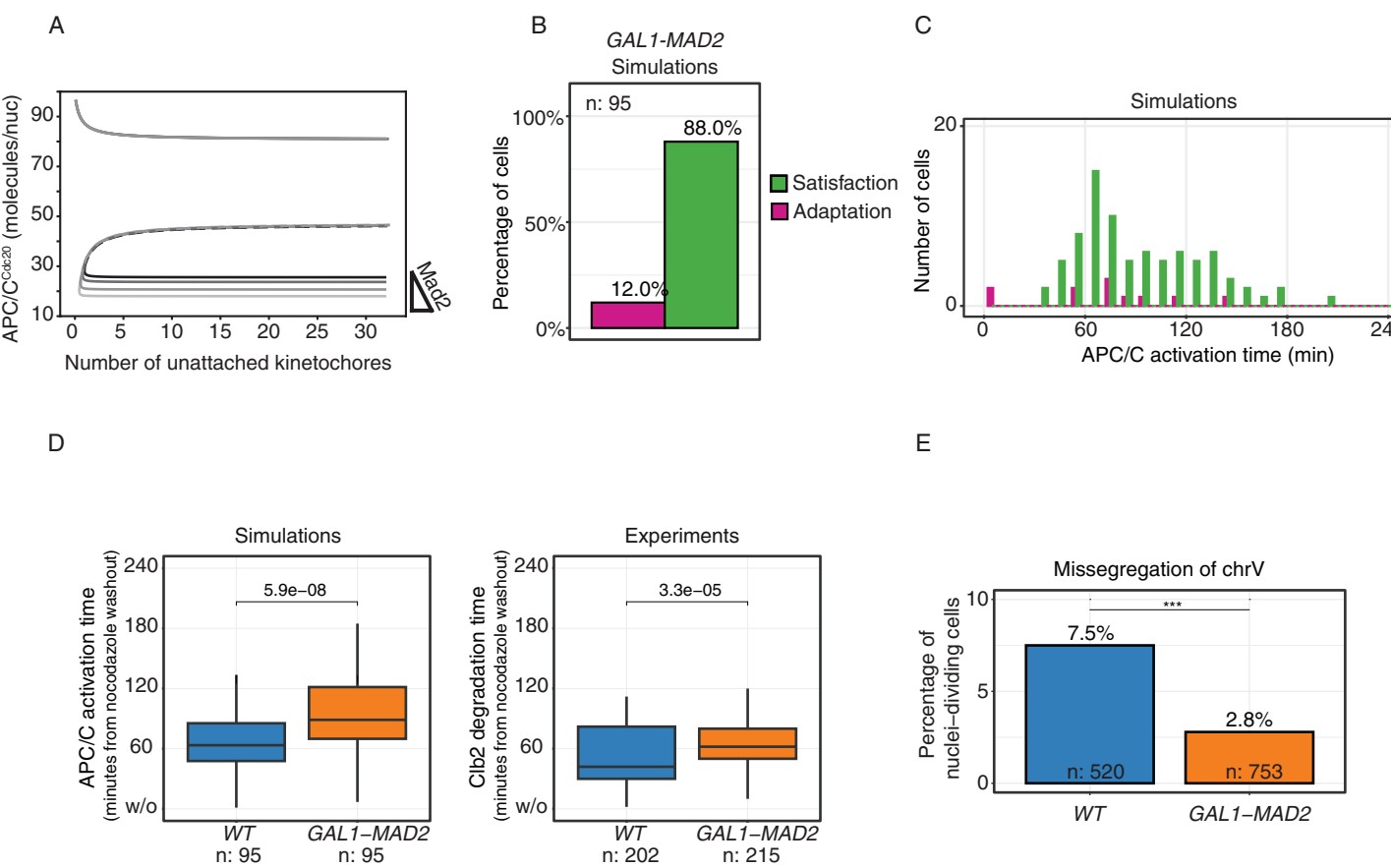

**Fig 5. Dynamics of nocodazole washout upon Mad2 overexpression. A)** Bifurcation diagrams showing the effect of increasing the Mad2 levels to 105%, 115% and 125% of Wild Type levels. Wild type in black. Equations in S1 Text, and parameters in Table 2. **B)** Barplots show percentages of simulated cells adapting or exiting, where adaptation is defined as APC/C activation with at least one unattached kinetochore. Reactions in Table 3, parameters in Table 2, initial conditions in Table 1 (checkpoint ON). **C)** Histogram of APC/C activation time in in simulated cells adapting (purple) or exiting (green). Details of simulations as in panel **(B)**. **D)** Comparison of Clb2 degradation time (experiments) and APC/C activatiom time (simulations) in wild-type and *GAL1-MAD2* cells. In the experiments (right), cells growing in YPRG are arrested in G1 and released in nocodazole for 240 minutes, when nocodazole is removed from the media. Cells carry Clb2-mCherry to record Clb2 degradation time. Only cells fully degrading Clb2 are included (see S2A Fig; 95% in wild-type, 87% in *GAL1-MAD2*). Simulations (left) produced as described in panel **B**. **E)** Percentage of cells that missegregate chromosome V. Cells growing in YPRG are arrested in G1 and released in nocodazole for 180 minutes, when nocodazole is removed from the media. Cells carry tetR-GFP/tetO construct to keep track of chrV segregation, and Htb2-mCherry to score nuclear division time. Cells that do not undergo nuclear division are excluded from this barplot (see S5F Fig). Missegregation is defined based on 5 frames (50 minutes) around nuclear division time. A cell is marked as missegregating chrV if after nuclear division either the two GFP dots end in the same cell or only one GFP dot is visible (see S5D and S5E Fig). For statistical comparison we used an exact Fisher test, where the contingency table has wild-type or mutant cells versus correct or wrong segregation of chrV (p-value = 1.3 10⁻⁴).

To validate these predictions, we performed again nocodazole washout experiments using microfluidic devices and yeast strains with mCherry-tagged Clb2 and GFP-tagged Mad2. The slower timing of entry into anaphase was confirmed experimentally in the *GAL1-MAD2* system by following the timing of Clb2 degradation in the population (Fig 5D, right). Unfortunately, it was not possible to confirm also the model's predictions about adaptation. In *GAL1-MAD2*, GFP-Mad2 expressed from the endogenous promoter is diluted by the overexpressed untagged Mad2. For this reason, we could not detect Mad2 localization via microscopy, and thus we could not directly track adaptation events.

We decided, instead, to investigate the consequences of adaptation on chromosome segregation. The expectation is that missegregation decreases when adaptation decreases. More precisely, we followed the redistribution of chromosome V by tagging this specific

chromosome with GFP using the lacO/LacI system (in S5C Fig an example of arrested cell; in S5D Fig a cell segregating properly; and in S5E Fig and S2 Movie a missegregating cell). We used the same microfluidic device and protocol used for the wash out experiments: cells were arrested in G1 in microfluidic devices for 2 hours, then released in nocodazole-containing medium for 3 hours after which nocodazole was washed out.

Data show that missegregation of chromosome V decreases significantly in *GAL1-MAD2* cells after wash-out, in agreement with the hypothesis that these cells also decrease the number of adaptation events (Fig 5E). Using the same experimental setup we also analyzed chromosome segregation in the APC-A mutant. We confirmed its problems in APC/C re-activation after washout (43.8% of cells not diving nuclei, S5F Fig) and confirmed the reduced segregation errors compared to wild-type (S5G Fig).

To further investigate the causal relationship between adaptation and chromosome misseg-regation, we asked whether they were quantitatively correlated in the two strains for which we could measure them using the same protocol: WT and APC-A. Interestingly, in a scatter plot, the two quantities were indeed correlated (S5H Fig). We further asked whether adapta-tion could be a predictor for missegregation. Assuming that all chromosomes missegregate independently with probability similar to that of chromosome V, we inferred the number of cells that missegregate at least one chromosome upon drug washout (that is: 1-(no missegre-gation)=$1-(1-p)^n$, with p=0.075 and n=16 where p is the missegregation rate of chr V and n the total number of chromosomes in haploid yeast). This number was similar to the observed rate of adaptation for both WT and APC-A, suggesting that adaptation can be a predictor for missegregation.

In summary, our data suggest that reinforcing the checkpoint by increasing Mad2 levels decreases the chance of cells missegregating upon drug washout, and that adaptation and missegregation are causally related.

## Simple toy-models with bistability also reproduce experimental adaptation dynamics

In yeast, the molecular mechanisms underlying the positive feedback loop(s) involving mitotic checkpoint and APC/C are not well understood. So far, we implemented the mod-ule shown in Fig 1A where 'X' (Mps1 in the previous simulations) is part of the positive feedback loop. However, we have not explored the behavior of the second network (Fig 1A right); and even for the one we investigated, one may envision different implementations where Mps1 in the role of 'X' is replaced by a different player, e.g., Cdk1/Clb2. Hence, we asked whether the behaviors described so far are specific of the particular network we chose, or rather are typical properties of models based on mitotic checkpoints that include positive feedback loops.

To address this point, we built simple models of the mitotic checkpoint based on the two classes of positive feedback loops in Fig 1A (S6A-S6G Fig), and we asked whether they could account for: (i) bistability in anaphase (as in Fig 1C and 1D), (ii) the observed adaptation dynamics during a prolonged arrest (Fig 2D and 2E) and (iii) the reduction in adapting cells in APC-A mutants compared to wild types (Figs 3C and 4C).

In both models, the positive feedback loops gave rise to bistable APC/C$^{Cdc20}$ activities (S6B-S6H Fig) that reproduced the behavior observed in [9] and simulated in Fig 1D (S6C-S6I Fig). When we used stochastic simulations (S6D-S6L Fig) we reproduced the exponential distribution of adaptation times similarly to what we observed experimentally (S6E-S6M Fig). We used these two models to reproduce also the washout experiments in WT and the APC-A mutant. Results confirmed the decrease in the number of cells undergoing adaptation in APC-A mutants (S6F-S6N Fig).

In summary, our simulations show that adaptation dynamics are not a specific feature of the detailed model, but are generic properties of models which include positive feedback loops in the mitotic checkpoint network.

## Discussion

Inspired by a model that describes the mitotic checkpoint as a bistable system, we investigated why and how cells slip through a mitotic arrest. In particular, we showed that washing out checkpoint-activating drugs favors adaptation. The model, supported by experiments, suggests that adapting cells escape from the arrest thanks to stochastic fluctuations that cross the boundary between the checkpoint ON and checkpoint OFF states. Such fluctuations become more and more disruptive as the number of unattached kinetochores decreases, as it happens when drugs are washed out and the process of attachment is delayed (up to one hour in our experiments). We then used the model to devise strategies to avoid adaptation, which we validated experimentally. We showed that similar computational results can be obtained with other models that include different positive feedback loops.

We argue that our results may be relevant for containing aneuploidy in cells treated with drugs that target mitosis, both in patient treatment and for cell synchronization protocols in the laboratories. Since, as we show, adaptation/slippage and missegregation are causally related, decreasing adaptation can help decreasing the emergence of aneuploid cells.

### The steady-state trap

Our results show that prolonged mitotic arrests are difficult to escape from without risking missegregation, both during treatment and after washout. It may be surprising that missegregation following an arrest is relatively easy, when during a regular cycle the frequency of missegregation is very low ($\sim 10^{-5}$ in budding yeast [22]). We propose this to be because during a regular cycle cells do not have sufficient time to reach the checkpoint ON steady state. In budding yeast, there are overlapping mechanisms that guarantee proper kinetochore-microtubule attachments. Microtubules and kinetochores are constantly attached throughout the cell cycle, except for the limited time when centromeric DNA is replicated [23,24]. Yeast cells separate spindle pole bodies before microtubule/kinetochore attachment and have an intrinsic geometric bias for kinetochore orientation following the first attachment [25]. Even if the bias is lost, wild type cells thanks to Sgo1 manage to attach chromosomes efficiently [25]. Indeed, the mitotic checkpoint is only transiently, if at all, activated during a regular yeast cell cycle [26]. Only when these safeguard mechanisms fail and unattached kinetochores persist, the checkpoint becomes fully engaged. It stalls cell division, but it struggles to keep APC/C$^{Cdc20}$ levels under control. When cells reach this point, avoiding adaptation is not simple, since spontaneous fluctuations can be sufficient to drive the system towards the checkpoint OFF state.

In our model for preventing adaptation it is necessary to decrease the impact of random fluctuations. This can be done by lowering APC/C$^{Cdc20}$ activity during an arrest, which can be achieved either by reinforcing the mitotic checkpoint, or by lowering the affinity of APC/C for Cdc20. We show that both approaches effectively decrease the probability of adaptation during washout. Neither strategy has a strong phenotype in the absence of the drug. However, after treatment with nocodazole, the APC/C mutant is more disruptive, with a consistent fraction cells that does not even manage to degrade Clb2 after drug washout. Overexpression of Mad2 has a milder effect.

### Mammals and yeast

In mammalian cells, release from a transient spindle-poison treatment results in merotelic attachment [27,28], a condition which leads to errors in chromosome segregation but is not

recognized by the mitotic checkpoint. In that context, the inhibition of APC/C was shown to decrease chromosome missegregation, granting cells extra time to fix merotelic attachment independently from checkpoint activation. Indeed, it was shown that APC/C inhibition delayed the time of anaphase [28].

In haploid budding yeast, we observe similar results: APC/C inhibition delays transition into anaphase and also largely rescues missegregation. However, our interpretation of the results is different. In budding yeast, where each kinetochore is attached to only one microtubule, merotelic attachments are not possible. Moreover, Mad2 overexpression -- to a level that by itself is unable to slow down cells in mitosis -- suffices to achieve the same result as APC/C inhibition. We hypothesize that it does so synergizing with the signal originating from unattached kinetochores. Hence, we propose that in budding yeast APC/C$^{Cdc20}$ inhibition allows the mitotic checkpoint to be functional even with few unattached kinetochores. In this view, it reduces missegregation by diminishing the disruptive role of stochastic fluctuations.

How much can our results be extended to mammalian cells? This is not obvious. Yeast are unicellular organisms for which adaptation can be preferable than being indefinitely arrested in mitosis. For multicellular mammals, a different 'social' logic may prevail. At the same time, the molecular players controlling chromosome segregation are very similar and conserved between yeast and mammals. Even more relevant, in both systems, stochasticity seem to play a relevant role during adaptation. In budding yeast, single cell analysis shows that during a regular cell cycle mitotic cyclin is stable until it is degraded extremely quickly [6]. The timing of degradation is quite variable among different cells. In mammalian cells, cyclin B is slowly degraded during an arrest, and quickly degraded just before cells enter anaphase [4,12]. Nevertheless, the duration of mitotic arrest and the rate of cyclin degradation are quite variable and seemingly stochastic in the cell population [7]. Possibly, similar fluctuations in APC/C$^{Cdc20}$, which do not elicit any evident effect in yeast, in mammalian cells lead to the partial degradation of cyclin B and to the progressive weakening of the checkpoint. It is then possible that a mechanism similar to the one we described takes place also in mammalian cells.

It will be important in the future to understand whether this is the case. If the results will be confirmed, they could help devising new strategies to improve the efficacy of microtubule targeting drugs by minimizing the rate of adaptation and missegregation during and after drug treatment.

## Methods

### Yeast strains and growth conditions

Every strain (S1 Table) derives from *W303* (*ade2-1, trp1-1, can1-100, leu2-3,112, his3-11,15, ura3*), or was backcrossed at least three times with it. Apart from FCCS experiments, cells grew at 30˚ in YP medium (1% yeast extract, 2% Bacto Peptone, 50 mg/l adenine) supplemented with either 2% glucose (YPD), or 2% raffinose (YPR), or 2% galactose + 2% raffinose (YPRG). To induce G1-arrest, cells were treated with 3 μg/ml α-factor for 2 hours. To induce mitotic arrest, cells were treated with 15 μg/ml of nocodazole (from a 100X stock in DMSO). In experiments with prolonged exposure to nocodazole, α-factor was added at high concentration (12 μg/ml) after 3 hours from G1-release into nocodazole, to prevent cells to enter a new S-phase after adaptation. In experiments with induced *GAL1*-promoter, cells were first grown in YPR, and galactose was added at 2% concentration one hour before the G1-release.

### Protein tagging and genetic alterations

To create strains for FCCS, we used the approach described in [3]. Briefly, to reduce autofluorescence, we replaced the mutated *ade2-1* gene by one-step gene replacement with functional wild-type *ADE2* gene. To tag the proteins we used three fluorophores in tandem to

increase the signal, and we used them monomeric and with different codon usage to reduce the chances of recombination (using plasmids from [30]). Using the S-primer strategy [31], we tagged Cdc23, Cdc16, Apc5, Mad2, Mad3 right before the STOP codon. We used the same plasmid and strategy to tag Clb2 with mCherry. We tested all the obtained strains for viability at different temperatures (23˚, 30˚, 37˚), and with Ethanol/Glycerol as carbon source to check mitochondrial functionality. Those in which Mad2 or Mad3 were tagged were successfully tested for SAC proficiency.

*GAL1-MAD2* constructs have been published previously[21]. APC-A strains are a kind gift from A. Murray [20]. Mad2-GFP used in single-cell experiments is a kind gift from T. Tanaka. The *TtetR/tetO* construct is a kind gift from S. Piatti.

### Single-cell microscopy

Single-cell analyses were performed as in [3] and is briefly described below.

### Cell growth

Cell grew overnight in YPD or YPR at 30˚. To arrest them in G1, cells were treated with α-factor in flasks while growing exponentially. In cells growing in YPR, galactose was added after one hour from α-factor addition to the final concentration of 2% (resulting in YPRG). Then cells were diluted, sonicated, and transferred to a microfluidic chamber (Y04C-02-5PK, CellASIC). Once perfused in the chamber, cells were trapped with a 5-minutes flow at 6 psi, and then shifted to 2 psi to complete the 2-hours arrest. G1-release was performed in the chamber, with a 5-minutes flow at 6 psi of fresh medium without pheromone. Then noco-dazole was perfused in the chamber. In washout experiments, after 180 minutes in nocodazole the drug was removed with a 5-minutes flow at 8 psi using medium supplemented with 1% DMSO. During every other growth-step, the flow was set to 2 psi.

The presence of Raffinose and Galactose as carbon source is essential to induce Mad2 expression from the *GAL1* promoter. We chose to use the same carbon sources also for wild-type or APC-A experiments, to keep the whole experimental set more consistent. Glucose (YPD) was used only in the experiments presented in Fig 2, to repeat what published in [3].

### Acquisition

Time-lapse movies were recorded using a DeltaVision Elite imaging system (Applied Precision) based on an inverted microscope (IX71; Olympus) with a camera (CoolSNAP HQ2; Photometrics) and a UPlanFL N 60x (1.25 NA) oil immersion objective lens (Olympus). Mad2-GFP was acquired using 11 Z-stacks spaced 0.3 μm, at 10% lamp power for 0.07 s. Clb2-mCherry was acquired with a single Z-stack at 10% lamp power for 0.5 s. tetR-GFP (for chromosome V detection) was acquired, starting 10 minutes before nocodazole removal (170 minutes from G1 release), using 8 Z-stacks spaced 0.55 μm at 32% lamp power for 0.20 s. Htb2-mCherry was acquired with a single Z-stack, at 32% lamp power for 0.15 s. The coex-istence of different acquisition settings was made possible by *ad hoc* scripting of the Delta-Vision acquisition procedure. The phototoxicity of the acquisition settings was measured by comparing the cell-cycle duration in excited and non-excited cells, for which we detected no significant difference. GFP and Cherry were acquired using single bandpass filters (EX475/28 EM523/36 and EX575/25 EM632/6, respectively).

### Cell segmentation and cell tracking

Cells were segmented and tracked analyzing phase-contrast images with a customized version of the software Phylocell (developed in Matlab by Gilles Charvin and colleagues [32] and

available on GitHub (https://github.com/gcharvin/phyloCell). Pixels in the segmented areas of mother and daughter cells were analyzed together until cells divided. For each fluorescence channel and each frame, we subtracted the background, computed as the mean value of the non-segmented area. Fluorescence signals were analyzed using custom software in Matlab.

## Image analysis – Mad2/Clb2

To identify Mad2 localization, Mad2-GFP images were deconvolved using SoftWoRx software, and projected using the maximum intensity projection. Mad2-GFP localization was determined, as in [3], using a Localization Index defined using the Laplacian of Gaussian (LoG) operator whose matrix is

-0.0085 0.0038 -0.0085
0.0038 0.0187 0.0038
-0.0085 0.0038 -0.0085

For each cell and time frame, we defined the maximum of the segmented and filtered image as Localization Index as shown below:

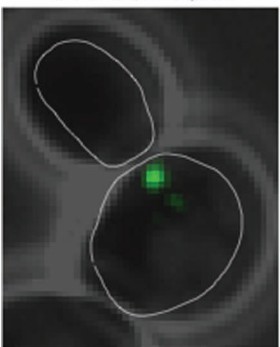

**REF and GFP signals**

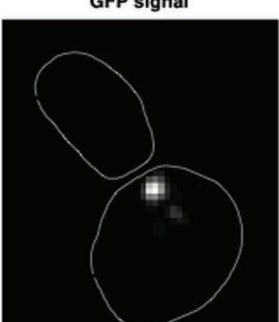

**GFP signal**

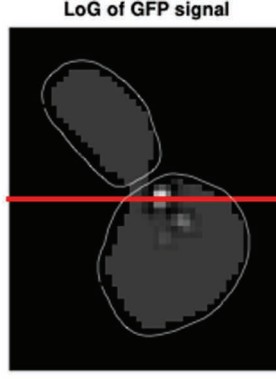

**LoG of GFP signal**

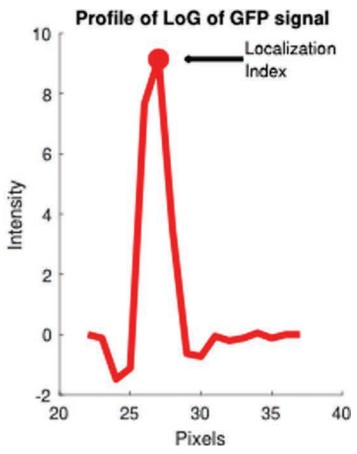

**Profile of LoG of GFP signal**

Mad2 is considered localized when its Localization Index lies above a threshold. The threshold is 110% of the maximum value of the Mad2 Localization Index measured in the first 20 minutes of the experiments, when cells are in G1.

To estimate the amount of Mad2-GFP at kinetochores, we estimated the total fluorescent signal at kinetochores. We localized the kinetochores as the peak value of the max projection of the deconvolved images, and then selected the 5x5 pixels around it as "kinetochore mask" as shown here:

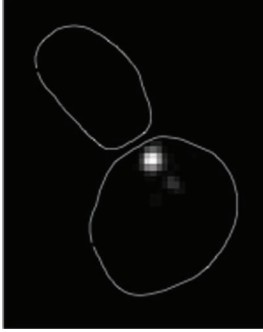

**Deconvolved signal**

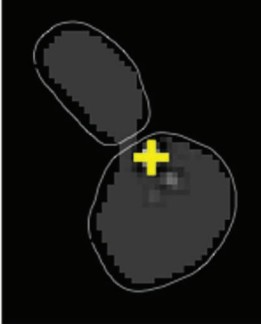

**LoG of Deconvolved signal and peak location**

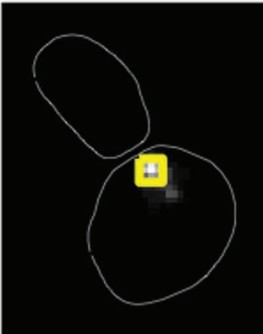

**Non-deconv signal and kinetochore mask**

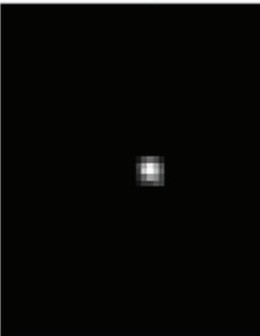

**Mask on non-deconv signal GFP signal**

To estimate the total fluorescence, we could not use the deconvolved images since deconvolution does not linearly scale with the total fluorescent signal. We then used the average projection of the non-deconvolved image. 'Mad2 value at kinetochores' is the sum of the fluorescence values in the 5x5 square in the average projection of non-deconvolved images.

Clb2-mCherry signal was averaged over the cells' pixels. It was then smoothed over time using a Savitsky-Golay filtering. Clb2 degradation time was identified as the time after the first budding when the smoothed Clb2 signal starts decreasing and falls. For sake of simplicity, we use the term 'Clb2 degradation time' rather than 'time of the beginning of Clb2 degradation', although the latter is more precise. Clb2 degradation rate was estimated by fitting Clb2 signal on 4 time-points starting from the one of Clb2 degradation. As a fitting function, we used an exponential. The exponent of the exponential, changing the sign from negative to positive, is the Clb2 degradation rate.

Cells with Clb2 degradation rate below 0.01 or degrading less than 50% of the total Clb2 in the 90 minutes following the degradation time are marked as "partially degrading Clb2".

## Image analysis – chrV/Histone

To identify nuclear division time, the Htb2-mCherry signal was analyzed using *k*-means algorithm. The nuclear-division frame was detected as the first frame when (i) two clusters were detected, and (ii) the distance between the two was at least 75% of the distance between the centroids of mother and daughter cells. If no such frame was detected but the cell or its daughter budded, the budding frame defines the division frame, as an approximation. If still no frames defined nuclear division, the cell was marked as 'arrested' (see S5C Fig). Automated detection was visually checked and corrected when needed.

To identify chromosome V segregation, we deconvolved tetR-GFP signal using Soft-WoRx software, and projected it using the maximum intensity projection. We then analysed five frames: from one before to three after nuclear division, as defined above. Bright spots due to tetR-GFP were detected with a Matlab script based on the FastPeakFind algorithm. A cell was marked as 'missegregating chromosome V' if either two bright spots ended up in the same cell or if, after nuclear division, only one bright spot was detected. Automated detection was visually checked and corrected when needed. See S5D and S5E Fig for examples.

## Fluorescence-Cross-Correlation Spectroscopy (FCCS)

FCCS was performed as in [3]. Briefly, cells were grown at 23C° in synthetic complete medium with glucose, supplemented with 1% Bacto Peptone to maximize the effect of nocodazole [33]. G1- and nocodazole-arrest were performed as for the single-cell experiments. Once sampled, cells were immobilized by pre-treating glass-bottomed wells with 0.1% Bioconext. Cells were measured when in metaphase, 150 to 210 minutes after release into nocodazole from G1. Only one measurement per cell was performed, lasting 45 seconds.

The measurements were acquired with a MicroTime 200 time-resolved confocal microscope. Cells were excited using a 485nm pulsed laser diode head (LDH-D-C-485, PicoQuant), pulsing at 20MHz and a 561nm CW laser (Cobolt Jive, Cobolt). The laser beam was positioned in the nucleus, identified by the signal from Mad2-GFP or Cdc23-mCherry. Brightfield images were used to check the correct phase of the cell cycle of the acquired cells.

Acquisition- and overlapping-volumes were determined using calibration dyes Atto-488 and Atto-565.

**Table 3. Reactions for stochastic simulations.**

| Name | Reaction | Rate Law |
|---|---|---|
| Cdc20 promoter activation | $p_{offCDC20} \rightarrow p_{onCDC20}$ | $k_{onCDC20} \cdot p_{offCDC20}$ |
| Cdc20 promoter inactivation | $p_{onCDC20} \rightarrow p_{offCDC20}$ | $k_{offCDC20} \cdot p_{onCDC20}$ |
| Cdc20 synthesis | $p_{onCDC20} \rightarrow C + p_{onCDC20}$ | $k_{synCDC20} \cdot p_{onCDC20}$ |
| Mps1 promoter activation | $p_{offX} \rightarrow p_{onX}$ | $k_{onX} \cdot p_{offX}$ |
| Mps1 promoter inactivation | $p_{onX} \rightarrow p_{offX}$ | $k_{offX} \cdot p_{onX}$ |
| Mps1 synthesis | $p_{onX} \rightarrow Mps1 + p_{onX}$ | $k_{synX} \cdot p_{onX}$ |
| Cdc20 background degradation | $C \rightarrow \varnothing$ | $k_{degBG} \cdot C$ |
| Mps1 degradation | $Mps1 \rightarrow \varnothing$ | $k_{degX} \cdot Mps1 \cdot AC$ |
| Mps1 background degradation | $Mps1 \rightarrow \varnothing$ | $k_{degBGX} \cdot Mps1$ |
| □□□ activation | $Mad_i \rightarrow Mad_a$ | $k_{act} \cdot X \cdot \dfrac{k' \cdot nUK}{J_n + nUK} \cdot \dfrac{Mad_i}{J + Mad_i}$ |
| □□□ inactivation | $Mad_a \rightarrow Mad_i$ | $k_{inact} \cdot Mad_a \cdot J + Mad_a$ |
| MCC (mitotic checkpoint complex) formation | $Mad_a + C \rightarrow MC$ | $k_{assMC} \cdot M \cdot C$ |
| MCC dissociation | $MC \rightarrow Mad_a + C$ | $k_{dissMC} \cdot MC$ |
| APC/C$^{Cdc20}$ (AC) formation | $A + C \rightarrow AC$ | $k_{assAC} \cdot AC$ |
| APC/C$^{Cdc20}$ dissociation | $AC \rightarrow A + C$ | $k_{dissAC} \cdot AC$ |
| APC/C$^{MCC}$ (ACMC) formation | $MC + AC \rightarrow ACMC$ | $k_{assACMC} \cdot MC \cdot AC$ |
| APC/C$^{MCC}$ dissociation | $ACMC \rightarrow MC + AC$ | $k_{dissACMC} \cdot ACMC$ |
| APC/C$^{MCC}$ degradation | $ACMC \rightarrow A + C + M$ | $k_{deg} \cdot ACMC$ |
| MCC background degradation | $MC \rightarrow M$ | $k_{degBG} \cdot MC$ |
| APC/C$^{Cdc20}$ background degradation | $AC \rightarrow A$ | $k_{degBG} \cdot AC$ |
| APC/C$^{MCC}$ background degradation | $ACMC \rightarrow A + M$ | $k_{degBG} \cdot ACMC$ |
| Kinetochore attachment dynamics | $nUK \rightarrow \varnothing$ | $nuk \cdot k_{att}$ |

Auto- and cross-correlation functions were computed and fitted using FluctoAnalyzer [34], correcting for background autofluorescence and green-to-red bleedthrough. The resulting functions were fitted to a two-components model, assuming triplet-like blinking state (see following equation).

$$G(\tau)=\left(1-\theta_T+\theta_T e^{-\frac{\tau}{\tau_T}}\right)\frac{1}{N}\left[\sum_{j=1}^{2}f_j\frac{1}{1+\frac{\tau}{\tau_{D,j}}}\sqrt{\frac{1}{1+\frac{1}{K^2}\frac{\tau}{\tau_{D,j}}}}\right]$$

The optical parameter κ was fitted using the calibration dyes. From this function we computed G(0), which is equal to 1/N, where N is the average number of fluorescent molecules in the confocal volume. We will refer to this value as $G_{rr}$ when studying the red autocorrelation function or $G_{gg}$ when we study the green. From this value we computed the absolute concentration of each fluorescent protein by using the following equation:

$$C_i=\frac{1}{V_i N_A G_{ii}}\qquad i=r,g$$

where $N_A$ is the Avogadro constant and $V_i$ is the acquisition volume for the *green* or *red* channel. To compute the concentration of the complex we used the following equation (see Ref [35], page 367):

$$C_{rg}=\frac{G_{rg}V_{rg}}{G_{rr}V_r N_A G_{gg}V_g}$$

$$=\frac{G_{rg}V_{rg}N_A}{G_{rr}V_r N_A G_{gg}V_g N_A}$$

$$=G_{rg}V_{rg}N_A C_r C_g$$

where the subscript *rg* represents parameters coming from the fitting of the cross-correlation function. $V_{rg}$ is the overlapping volume.

The database containing the fitted parameters was analyzed using an automatized pipeline written using RStudio [36]. We identified as unreliable (e.g.,: trembling cells, laser beam too close to the outer membrane) and removed all measurements with $R^2 < 0.99$ for one of the two proteins (~5% of the measurements). Similarly, measurements with $R^2 < 0.3$ for the cross-correlation fitting were removed (<0.1%). The reason why we used a lower quality threshold for the complex is to avoid removing cells where the two proteins are not interacting.

### Plotting and simple statistical tests

Plots were created using the `ggplot` library in R. Violin plots use a Gaussian kernel with bandwidth adjustment equals 2. In the boxplots, the box spans the interquartile range (IQR, from the 25th to the 75th percentiles), and the central band represents the median. The lower (upper) whisker extends from the box to the smallest (largest) value no further than 1.5*IQR from the box. When boxplots and violin plots are compared, we used unpaired Wilcoxon test.

### Supporting information

**S1 Fig. Bifurcation analysis of bistability in the mitotic checkpoint network.** From left to right, bifurcation diagram for three consecutive moments of the experiment simulated in Fig 1D. At the beginning of the experiment (left), cells are in the checkpoint OFF state, arrested in anaphase (by the *cdc15-2* mutation). Although all kinetochores are attached, on the x-axis the

checkpoint activating signal is not zero due to the lack of tension between sister chromatids. When Mps1 is overexpressed (middle), the checkpoint OFF state is greatly reduced, and the system is attracted to the only available steady state, checkpoint ON. After the overexpression stops (right), the original checkpoint OFF steady state is available again. Yet, cells remain attracted to the checkpoint ON steady state.
(PDF)

**S2 Fig. Examples of adapting and exiting cells. A)** Classification of cell behaviors based on Clb2 dynamics. From left to right, a cell defined as arrested, where Clb2 signal is never degraded; a cell partially degrading Clb2; and a cell fully degrading Clb2. Rightmost panel: barplots presenting the percentages of the different classes of cells in wild-type cells arrested in nocodazole indefinitely. **B)** Two examples of Mad2 Localization Index (black curve) and Clb2 mean signal (grey curve) over time. In the left panel, a cell defined as exiting: at the moment of Clb2 degradation (grey dot, dashed vertical line) Mad2 Localization Index lies below the threshold (dashed horizontal line). The threshold is defined based on the Localization Index value in G1. The green dot highlights that the value of Mad2 Localization Index at Clb2 degradation time is below the threshold. In the right panel, a cell defined as adapting. In this cell, Mad2 Localization value at the Clb2 degradation time is above the threshold. This is highlighted by the purple dot. These examples come from a washout experiment but the definition applies also to cells always kept in nocodazole. Details of the analysis are presented in the Image analysis section of the Materials and Methods.
(PDF)

**S3 Fig. Exit and adaptation with few or many unattached kinetochores. A)** Barplot presenting the percentages of the different classes of Clb2 behavior in wild-type cells arrested in nocodazole for 180 minutes. For the definition of the cell classes see S2A Fig. **B)** Comparison between experimental and simulated values for kinetochores-attachment dynamics. Experimental data show the dynamics of cells with Mad2 Localization Index above the threshold over. Simulated data show the dynamics of unattached kinetochores (percentage of the number of unattached kinetochores per cell). **C)** Bifurcation diagrams with APC/C$^{Cdc20}$ at steady state in function of the number of unattached kinetochores superimposed with the trajectories of attachment (Fig 3B). Representative trajectories are shown for exit (left), and adaptation with few (center) or many (right) unattached kinetochores.
(PDF)

**S4 Fig. Adaptation in APC-A mutants. A)** Bifurcation analysis shows how increasing the distance between stable (checkpoint ON) and unstable steady states minimizes the effects of random fluctuations. **B)** Absolute concentrations of APC/C$^{MCC}$ and total MCC as measured by Fluorescence Cross-Correlation Spectroscopy (FCCS). Wild-type or APC-A cells are arrested in G1 and released in nocodazole. Cells are measured in nocodazole between 150 and 210 minutes. To measure APC/C$^{MCC}$, we measured the cross-correlation of Mad2-GFP and Cdc23-mCherry, while for total MCC we used Mad2-GFP and Mad3-mCherry. To compare the distribution of concentrations we used a linear model adjusting for batch effects for experiments performed on different days. p-values are 0.81 for APC/C$^{MCC}$ and 2.7 10$^{-6}$ for MCC total. Details of treatment, measurements, and analyses are presented in the FCCS section of Materials and Methods. **C)** Left panel: barplot presenting the percentages of the different classes of Clb2 behavior in APC-A cells arrested in nocodazole indefinitely. For the definition of the different classes see S2A Fig. Central panel: cumulative distribution of length of mitotic arrest in wild-type or APC-A cells. Wild-type curve is the same presented in Fig 2B. Cells growing in YPD are arrested in G1 and released in nocodazole, indefinitely. Cells carry Mad2-GFP to record checkpoint activation status, and Clb2-mCherry to record both mitotic

entry (Clb2 rise time) and mitotic exit (Clb2 degradation time). The duration of mitotic arrest is defined as the difference between mitotic exit and mitotic entry. Right panel: cumulative distribution of APC/C activation time for the simulated data. Adaptation is defined as APC/C crossing the activation threshold with at least one unattached kinetochore. Reactions in Table 3, parameters in Table 2, initial conditions in Table 1 (checkpoint ON). **D)** Barplot presenting the percentages of the different classes of Clb2 behavior in APC-A cells arrested in nocodazole for 180 minutes. For the definition of the cell classes see S2A Fig. **E)** Bifurcation diagrams of wild type cells (same as in Fig 1C) and APC-A mutant, where the association rate between APC/C and Cdc20 was decreased to 65% of the WT. Here, the saddle node is located at 0.3 unattached kinetochores. Equations in S1 Text, and parameters in Table 2.
(PDF)

**S5 Fig. Adaptation in GAL1-MAD2. A)** Left panel: Cumulative distribution of length of mitotic arrest in wild-type or Mad2-overepressing cells. Cells growing in YPRG are arrested in G1 and released in nocodazole, indefinitely. Cells carry Mad2-GFP to record checkpoint activation status, and Clb2-mCherry to record both mitotic entry (Clb2 rise time) and mitotic exit (Clb2 degradation time). Mitotic arrest is defined as the difference between mitotic exit and mitotic entry. Central panel: barplots with the percentages of the different classes of Clb2 behavior in wild-type or *GAL1-Mad2* cells arrested indefinitely in nocodazole. For the definition of the cell classes see S2A Fig. Right panel: cumulative distribution of APC/C activation time for the simulated data. Adaptation is defined as APC/C crossing the activation threshold with at least one unattached kinetochore. Reactions in Table 3, parameters in Table 2, initial conditions in Table 1 (checkpoint ON). **B)** Bifurcation diagrams of wild type cells (same as in Fig 1C) and *GAL1-MAD2* mutant. Expression level of Mad2 is increased to 125% compare to WT. In *GAL1-MAD2*, the saddle node bifurcation is located at 0.4 unattached kinetochores. Equations in S1 Text, and parameters in Table 2. **C)** Example of an arrested cell taken from a movie of cells growing in YPRG, arrested in G1 and released in nocodazole for 180 minutes, when nocodazole is removed from the media. Cells carry the tetR-GFP/tetO construct to keep track of chrV segregation, and Htb2-mCherry to score nuclear division time. Definition of nuclear division time is presented in the Image analysis section of the Materials and Methods. **D)** Example of a cell correctly segregating chrV. In the lowest row, yellow and orange traces are cells segmentations, while red crosses mark the position of chrV. In the second column is the frame when nuclear division is registered. Definitions of correct or wrong segregation are presented in the Image analysis section of the Materials and Methods. **E)** Example of a cell missegregating chrV. Details in panel (**C**). **F)** Percentage of cells that divide nuclei. Cells growing in YPRG are arrested in G1 and released in nocodazole for 180 minutes, when nocodazole is removed from the media. Cells carry the tetR-GFP/tetO construct to keep track of chrV segregation, and Htb2-mCherry to score nuclear division time. **G)** Percentage of cells that missegregate chromosome V. Here, 100% means the cells that divide nuclei as shown in panel **F**. Data for wild-type and *GAL1-MAD2* cells are presented also in Fig 5E. For statistical comparisons we used an exact Fisher test, where the contingency table has wild-type or mutant cells versus correct or wrong segregation of chrV (p-values: $5 \cdot 10^{-3}$ for APC-A, $1.3 \cdot 10^{-4}$ for GAL-MAD2). **H)** Scatter plot of missegregation rate (from panel G) and adaptation rate (Fig 3A for WT and Fig 4B for APC-A). The red curve represents the probability P of having at least one missegregation event. It is computed as $1-(1-p)^{16}$, where p is the probability of missegregating one chromosome, under the assumption that all chromosomes share the same probability of missegregation.
(PDF)

**S6 Fig. Analysis of simple models. A)** Wiring diagram of simple Model 1, which belongs to the class shown in Fig 1A left. **B)** The model shows bistability. It also reproduces the alteration

in the bifurcation diagram of APC/C similarly to what seen for the detailed model in Fig 4A. **C)** Simulation of the Palframan's experiment[9] to show bistability in the simple model, in analogy with Fig 1D. **D)** Stochastic simulations with fixed number of unattached kinetochores give rise to the exponential distribution shown in **(E)**, in analogy with Fig 2D and 2E. **F)** APC-A mutants decrease the frequency of adaptations upon washout, similarly to Fig 4C. **G)** Wiring diagram of simple Model 2, which belongs to the class shown in Fig 1A right. **H)** The model shows bistability. It also reproduces the alteration in the bifurcation diagram of APC/C similarly to what seen for the detailed model in Fig 4A. **I)** Simulation of the Palframan's experiment [9] to show bistability in the simple model, in analogy with Fig 1D. **L)** Stochastic simulations with fixed number of unattached kinetochores give rise to the exponential distribution shown in **(M)**, in analogy with Fig 2D and 2E. **N)** APC-A mutants decrease the frequency of adaptations upon washout, similarly to Fig 4C.
(PDF)

**S1 Movie.  An example of a cell satisfying the checkpoint upon nocodazole washout.**
(AVI)

**S2 Movie.  An example of an adapting cell upon nocodazole washout.**
(AVI)

**S1 Table.  List of strains used in this study.**
(PDF)

**S1 Text.  Model's description** .
(DOCX)

## Acknowledgments

We thank all members of the Ciliberto lab, Silke Hauf and Attila Becskei for discussions. We received: APC-A strains from Andrew Murray (Harvard University); Mad2-GFP from Tomo Tanaka (University of Dundee); tetR/tetO construct from Simonetta Piatti (CNRS Montpellier). The original idea of adaptation as a stochastic transition in a bistable system we owe to Béla Novák.

## Author contributions

**Conceptualization:** Paolo Bonaiuti, Fridolin Gross, Andrea Ciliberto.

**Data curation:** Alma Beatrix Stier, Paolo Bonaiuti, Fridolin Gross.

**Formal analysis:** Paolo Bonaiuti.

**Funding acquisition:** Andrea Ciliberto.

**Investigation:** Alma Beatrix Stier, Paolo Bonaiuti, Fridolin Gross, Andrea Ciliberto.

**Methodology:** Alma Beatrix Stier, Paolo Bonaiuti, János Juhász, Fridolin Gross.

**Project administration:** Andrea Ciliberto.

**Resources:** Andrea Ciliberto.

**Software:** Alma Beatrix Stier, Fridolin Gross, Andrea Ciliberto.

**Supervision:** János Juhász, Fridolin Gross, Andrea Ciliberto.

**Validation:** Paolo Bonaiuti.

**Visualization:** Alma Beatrix Stier, Paolo Bonaiuti, Andrea Ciliberto.

**Writing – original draft:** Andrea Ciliberto.

**Writing – review & editing:** Paolo Bonaiuti, Fridolin Gross.

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
