## [Decision Letter · Decision Letter 0]

4 Dec 2024

PCOMPBIOL-D-24-01748

Increased risk of slippage upon disengagement of the mitotic checkpoint

PLOS Computational Biology

Dear Dr. Ciliberto,

Thank you for submitting your manuscript to PLOS Computational Biology. After careful consideration, we feel that it has merit but does not fully meet PLOS Computational Biology's publication criteria as it currently stands. Therefore, we invite you to submit a revised version of the manuscript that addresses the points raised during the review process.

Please submit your revised manuscript within 30 days Feb 03 2025 11:59PM. If you will need more time than this to complete your revisions, please reply to this message or contact the journal office at ploscompbiol@plos.org. Please include the following items when submitting your revised manuscript:

We look forward to receiving your revised manuscript.

Kind regards,

James R. Faeder

Academic Editor

PLOS Computational Biology

Jason Haugh

Section Editor

PLOS Computational Biology

Feilim Mac Gabhann

Editor-in-Chief

PLOS Computational Biology

Jason Papin

Editor-in-Chief

PLOS Computational Biology

**Journal Requirements:**

At this stage, the following Authors/Authors require contributions: Beatrix Stier, Paolo Bonaiuti, Janos Juhasz, Fridolin Gross, and Andrea Ciliberto. Please ensure that the full contributions of each author are acknowledged in the "Add/Edit/Remove Authors" section of our submission form.

4) We noticed that you used the phrase 'not shown' in the manuscript. We do not allow these references, as the PLOS data access policy requires that all data be either published with the manuscript or made available in a publicly accessible database. Please amend the supplementary material to include the referenced data or remove the references.

5) Your manuscript is missing the following sections: Methods. Please ensure all required sections are present and in the correct order. Make sure section heading levels are clearly indicated in the manuscript text, and limit sub-sections to 3 heading levels. An outline of the required sections can be consulted in our submission guidelines here: 

6) Please upload all main figures as separate Figure files in .tif or .eps format. For more information about how to convert and format your figure files please see our guidelines: 

7) We have noticed that you have uploaded Supporting Information files, but you have not included a list of legends. Please add a full list of legends for your Supporting Information files after the references list.

8) We notice that your supplementary Figures, Tables, and information are included in the manuscript file. Please remove them and upload them with the file type 'Supporting Information'. Please ensure that each Supporting Information file has a legend listed in the manuscript after the references list.

9) Thank you for stating that "All data (code) will be available." We strongly recommend all authors decide on a data sharing plan before acceptance, as the process can be lengthy and hold up publication timelines. Please note that, though access restrictions are acceptable now, your entire data will need to be made freely accessible if your manuscript is accepted for publication. This policy applies to all data except where public deposition would breach compliance with the protocol approved by your research ethics board. If you are unable to adhere to our open data policy, please kindly revise your statement to explain your reasoning and we will seek the editor's input on an exemption. Please be assured that, once you have provided your new statement, the assessment of your exemption will not hold up the peer review process.

10) Please amend your detailed Financial Disclosure statement. This is published with the article. It must therefore be completed in full sentences and contain the exact wording you wish to be published.

2) State what role the funders took in the study. If the funders had no role in your study, please state: "The funders had no role in study design, data collection and analysis, decision to publish, or preparation of the manuscript.".

If you did not receive any funding for this study, please simply state: The authors received no specific funding for this work.

**Reviewers' comments:**

Reviewer's Responses to Questions

**Comments to the Authors:**

**Please note that the review is uploaded as an attachment.**

Reviewer #1: See attachment

Reviewer #2: In their article, Stier, Bonaiuti et al describe a mathematical model that models the mitotic checkpoint in yeast and gives a rationale how cells exit the checkpoint after prolonged misattachment of microtubli.

The model is a network model, is build upon a previous model where they added components for the checkpoint "sensor" and consists of a positive feedback and is formulated as ordinary differential equations. A deterministic analysis shows that the dynamics is bistable. A stochastic analysis of the model shows an escape kinetics that is similar to the kinetics observed in experiments (Fig 2B+E), although there is some difference (a delay in the experimental data.

They test the model with specific experiments. For instance, they compare how many cells escape the checkpoint if the checkpoint is only temporarily activated, or they perform overexpression experiments to proof the bistability experimetnally.

Overall I am very enthusiastic about this paper. It is a carfuly created model for a very relevant process that is very well supported by experimental data and really answers the question why cells escape the mitotic checkpoint. The experiments have been well thought through and I am particularly enthusiastic how the authors provide experimental evidence for bistability.

I have only very minor comments:

1) I am curious how the authors explain the delay in escape in experimental data (Fig. 2B). I assume that if you need to overcome a bareer in a bistable system by stochastic fluctuations, this would result in an exponental escape distribution (something similar you see in Fig. 2E). Is there a delay in when escape can be detected?

2) In parts the experimens are difficult to follow for compuational people. For instance, the nocodazole washout experiment would not be understandable for people who don't know the compound. Please go through the article and explain better what these experiments test and what the molecular consequence of e.g. nocodazole washout is.

Reviewer #3: Review of Ciliberto 2024

Disclaimer: I am not qualified to assess the computational model, so my comments are focused on the experimental sections.

This study investigates the mechanisms underlying mitotic checkpoint adaptation (or “slippage”) and its relationship to chromosome missegregation and aneuploidy. Using a combination of mathematical modelling and experimental validation in budding yeast, the authors demonstrate first, that slippage is stochastic, driven by fluctuations in APC/C-Cdc20 activity, and second, that chromosome missegregation is exacerbated during or after nocodazole washout. The paper identifies two potential interventions to mitigate slippage by weakening the checkpoint -- with APC mutants or Mad2 overexpression. These insights may be relevant for enhancing the efficacy of microtubule-targeting drugs in therapeutic contexts. However, comparisons with mammalian cells are difficult given important differences between the two systems.

The study combines predictive modelling with detailed experimental validation, which is a powerful combination that strengthens the robustness of its findings. The paper introduces a bistability-driven framework for understanding mitotic checkpoint adaptation, providing a new perspective on mechanisms generating aneuploidy. In particular, the observation that nocodazole washout significantly increases the likelihood of chromosome missegregation and aneuploidy is particularly compelling, and in my opinion could be given more weight.

I recommend addressing the following issues that may significantly improve the paper before publication.

1. My main criticism is that I found the manuscript’s results difficult to follow because related experiments are dispersed across multiple figures (e.g., wild-type and mutant data for adaptation are split between Figures 3A, 4B, and 5D). Consolidating these into one or two cohesive figure(s) or section would clarify the message and improve readability.

2. The finding that cells are significantly more prone to chromosome missegregation after nocodazole washout seems to me like a novel and important insight. This result, shown in Figure 3A and related panels in other figures, should be highlighted more prominently, as it provides a clear link between checkpoint adaptation and aneuploidy. It should also be documented more clearly. For example, it would be nice to include in the main figure images of representative cells undergoing exit and adaptation, and mention the protocol for nocodazole washout (using microfluidics) in the main text.

3. Related to this, Figure 3A shows that 60% of cells adapt upon nocodazole washout, based on the observation that Clb2 degradation starts in the presence of Mad2 clusters, which should correspond to unattached kinetochores. Moreover, the model predicts that the time of adaptation for each cell is inversely correlated with its number of unattached kinetochores. However, this correlation is much lower for experiments than for the models, so I find the use of Mad2 intensity in Figure 3C as a proxy for unattached kinetochores unconvincing. Is the correlation driven mainly by the three data points with high Clb2 degradation times in graph 3C right? Much more compelling evidence for missegregation during adaptation is provided for chromosome V in Figures 5E and S5G, but requires confirmation that protocols are consistent across these experiments.

4. The proposed model suggests that fluctuations in APC/C-Cdc20 activity destabilise the checkpoint. Could these fluctuations be buffered by increasing ploidy, thereby reducing noise? Exploring or at least discussing this possibility might add depth to the paper. If this were the case, would the same noise exist in mammalian systems? A brief discussion on the role of noise and stochastic processes in cell fate decisions between the two systems would strengthen the manuscript.

5. In the introduction, the manuscript states, “After adaptation, chances are that cells missegregate chromosomes.” If this observation has been documented previously, a reference should be included. If it is speculation or a finding of the current study, consider moving this statement to the discussion section for clarity.

6. Please elaborate on possible mechanisms by which Mps1 overexpression activates the checkpoint in anaphase. This would help contextualise the role of Mps1 in the positive feedback loop.

7. In the introduction, define “PFL” (I assume it stands for Positive Feedback Loop) for clarity, as it may not be immediately familiar to all readers.

8. The phrase “cells are trapped in a condition that favours entry in anaphase with unattached kinetochores, even with the drug is washed out” should be revised. Instead of “even”, consider using “especially” or *“even more” to accurately reflect that washing out nocodazole increases the frequency of missegregation.

9. Provide more detail about the APC-A mutant. Specifically, which subunits are mutated and at which residues?

In conclusion, this manuscript presents a novel perspective on mitotic checkpoint adaptation, particularly highlighting the increased risk of chromosome missegregation during nocodazole washout. While the work is scientifically robust and overall of excellent quality, greater emphasis on the key findings and better organisation of the results would significantly enhance its impact. Despite some presentation issues, this study provides interesting new insights into the dynamics of checkpoint adaptation and its link to aneuploidy.

**Have the authors made all data and (if applicable) computational code underlying the findings in their manuscript fully available?**

Reviewer #1: **No: ** They have not made their computational code available. They should be required to do so.

Reviewer #2: Yes

Reviewer #3: Yes

PLOS authors have the option to publish the peer review history of their article (what does this mean? ). If published, this will include your full peer review and any attached files.

**Do you want your identity to be public for this peer review?** For information about this choice, including consent withdrawal, please see our Privacy Policy .

Reviewer #1: No

Reviewer #2: No

Reviewer #3: No

**Figure resubmission:**
---

## [Editor Report · Decision Letter 1]

14 Feb 2025

Dear Dr. Ciliberto,

We are pleased to inform you that your manuscript 'Increased risk of slippage upon disengagement of the mitotic checkpoint' has been provisionally accepted for publication in PLOS Computational Biology.

Best regards,

Jason M. Haugh

Section Editor

PLOS Computational Biology

---

## [Editor Report · Acceptance letter]

PCOMPBIOL-D-24-01748R1

Increased risk of slippage upon disengagement of the mitotic checkpoint

Dear Dr Ciliberto,

I am pleased to inform you that your manuscript has been formally accepted for publication in PLOS Computational Biology. Your manuscript is now with our production department and you will be notified of the publication date in due course.

With kind regards,

Lilla Horvath
